# Continuous DR-submodular Maximization: Structure and Algorithms

**An Bian**
ETH Zurich
ybian@inf.ethz.ch

**Kfir Y. Levy**
ETH Zurich
yehuda.levy@inf.ethz.ch

**Andreas Krause**
ETH Zurich
krausea@ethz.ch

**Joachim M. Buhmann**
ETH Zurich
jbuhmann@inf.ethz.ch

## Abstract

DR-submodular continuous functions are important objectives with wide real-world applications spanning MAP inference in determinantal point processes (DPPs), and mean-field inference for probabilistic submodular models, amongst others. DR-submodularity captures a subclass of non-convex functions that enables both exact minimization and approximate maximization in polynomial time.

In this work we study the problem of maximizing *non-monotone* continuous DR-submodular functions under general down-closed convex constraints. We start by investigating geometric properties that underlie such objectives, e.g., a strong relation between (approximately) stationary points and global optimum is proved. These properties are then used to devise two optimization algorithms with provable guarantees. Concretely, we first devise a "two-phase" algorithm with $1/4$ approximation guarantee. This algorithm allows the use of existing methods for finding (approximately) stationary points as a subroutine, thus, harnessing recent progress in non-convex optimization. Then we present a non-monotone FRANK-WOLFE variant with $1/e$ approximation guarantee and sublinear convergence rate. Finally, we extend our approach to a broader class of generalized DR-submodular continuous functions, which captures a wider spectrum of applications. Our theoretical findings are validated on synthetic and real-world problem instances.

## 1 Introduction

Submodularity is classically most well known for set function optimization, where it enables efficient minimization [23] and approximate maximization [31; 25] in polynomial time. Submodularity has recently been studied on the integer lattice [34; 33] and on continuous domains [3; 4; 36; 21], with significant theoretical results and practical applications. For set functions, it is well known that submodularity is equivalent to the diminishing returns (DR) property. However, this does not hold for integer-lattice functions or continuous functions, where the DR property defines a subclass of submodular functions, called DR-submodular functions.

In continuous domains, applying convex optimization techniques enables efficient minimization of submodular continuous functions [3; 36] (despite the non-convex nature of such objectives). In [4] it is further shown that continuous submodularity enables constant-factor approximation schemes for constrained monotone DR-submodular maximization and "box" constrained non-monotone submodular maximization problems.

Many real-world non-convex problems, such as maximizing the softmax extension of DPPs, require maximizing a *non-monotone* DR-submodular function over a *general* down-closed convex constraint. Yet, current theory [3; 4; 36] does not apply to this general problem setting, which motivates us to develop guaranteed and efficient algorithms for such problems.

Exploring the structure that underlies DR-submodularity is crucial to deriving guaranteed algorithms. Combined with a notion of non-stationarity for constrained optimization problems and a new notion of "strong DR-submodularity", we find a rich structure in the problem of continuous DR-submodular maximization. This in turn gives rise to two approximation algorithms with provable guarantees. Specifically, we make the following contributions:

- We bound the difference between objective values of stationary points and the global optimum. Our analysis shows that the bound is even tighter if the objective is strongly DR-submodular (see Definition 3).
- Based on the geometric properties, we present two algorithms: (i) A two-phase FRANK-WOLFE-style algorithm with $1/4$ approximation guarantee converges with a $1/\sqrt{k}$ rate; (ii) a non-monotone FRANK-WOLFE variant exhibits a $1/e$ approximation guarantee and converges sublinearly. Even though the worst-case guarantee of the first one is worse than the second, it yields several practical advantages, which we discuss in Section 4.2.
- We investigate a generalized class of submodular functions on "conic" lattices. This allows us to model a larger class of non-trivial applications. These include logistic regression with a non-convex separable regularizer, non-negative PCA, etc. To optimize them, we provide a reduction that enables to invoke algorithms for continuous submodular optimization problems.
- We experimentally demonstrate the applicability of our methods on both synthetic and real-world problem instances.

## 1.1 Problem Statement

**Notation.** We use boldface letters, e.g., $\boldsymbol{x}$ to represent a vector, boldface capital letters, e.g., $\mathbf{A}$ to denote a matrix. $x_i$ is the $i^{\text{th}}$ entry of $\boldsymbol{x}$, $A_{ij}$ is the $(ij)^{\text{th}}$ entry of $\mathbf{A}$. We use $\boldsymbol{e}_i$ to denote the standard $i^{\text{th}}$ basis vector. $f(\cdot)$ is used to denote a continuous function, and $F(\cdot)$ to represent a set function. $[n] := \{1, ..., n\}$ for an integer $n \geq 1$. $\|\cdot\|$ means the Euclidean norm by default. Given two vectors $\boldsymbol{x}, \boldsymbol{y}$, $\boldsymbol{x} \leq \boldsymbol{y}$ means $x_i \leq y_i, \forall i$. $\boldsymbol{x} \vee \boldsymbol{y}$ and $\boldsymbol{x} \wedge \boldsymbol{y}$ denote coordinate-wise maximum and coordinate-wise minimum, respectively.

The general setup of constrained non-monotone DR-submodular (see Definition 1 below) maximization is,

$$\max_{\boldsymbol{x} \in \mathcal{P}} f(\boldsymbol{x}), \tag{P}$$

where $f : \mathcal{X} \to \mathbb{R}$ is continuous DR-submodular, $\mathcal{X} = \prod_{i=1}^{n} \mathcal{X}_i$, each $\mathcal{X}_i$ is an interval [3; 4]. Wlog[1], we assume that the lower bound $\underline{\boldsymbol{u}}$ of $\mathcal{X}$ is 0, i.e., $\mathcal{X} = [0, \bar{\boldsymbol{u}}]$. The set $\mathcal{P} \subseteq [0, \bar{\boldsymbol{u}}]$ is assumed to be a down-closed convex set, where down-closedness means: $\boldsymbol{x} \in \mathcal{P}$ and $0 \leq \boldsymbol{y} \leq \boldsymbol{x}$ implies that $\boldsymbol{y} \in \mathcal{P}$. The diameter of $\mathcal{P}$ is $D := \max_{\boldsymbol{x}, \boldsymbol{y} \in \mathcal{P}} \|\boldsymbol{x} - \boldsymbol{y}\|$, and it holds that $D \leq \|\bar{\boldsymbol{u}}\|$. We use $\boldsymbol{x}^*$ to denote the global maximum of (P). One can assume $f$ is non-negative over $\mathcal{X}$, since otherwise one just needs to find a lower bound for the minimum function value of $f$ over $[0, \bar{\boldsymbol{u}}]$ (and box-constrained submodular minimization can be solved to arbitrary precision in polynomial time [3]). Over continuous domains, a DR-submodular function [4] is a submodular function with the diminishing returns (DR) property,

**Definition 1** (DR-submodular & DR property). *A function $f : \mathcal{X} \mapsto \mathbb{R}$ is DR-submodular (has the DR property) if $\forall \boldsymbol{a} \leq \boldsymbol{b} \in \mathcal{X}$, $\forall i \in [n], \forall k \in \mathbb{R}_+$ s.t. $(k\boldsymbol{e}_i + \boldsymbol{a})$ and $(k\boldsymbol{e}_i + \boldsymbol{b})$ are still in $\mathcal{X}$, it holds,*

$$f(k\boldsymbol{e}_i + \boldsymbol{a}) - f(\boldsymbol{a}) \geq f(k\boldsymbol{e}_i + \boldsymbol{b}) - f(\boldsymbol{b}). \tag{1}$$

If $f$ is differentiable, one can show that Definition 1 is equivalent to $\nabla f$ being an antitone mapping from $\mathbb{R}^n$ to $\mathbb{R}^n$. Furthermore, if $f$ is twice-differentiable, the DR property is equivalent to all of the entries of its Hessian being non-positive, i.e., $\nabla_{ij}^2 f(\boldsymbol{x}) \leq 0, \forall \boldsymbol{x} \in \mathcal{X}, i, j \in [n]$. A function $f : \mathcal{X} \mapsto \mathbb{R}$ is DR-supermodular iff $-f$ is DR-submodular. We also assume that $f$ has Lipschitz gradients,

**Definition 2.** *A function $f$ has L-Lipschitz gradients if for all $\boldsymbol{x}, \boldsymbol{y} \in \mathcal{X}$ it holds that,*

$$\|\nabla f(\boldsymbol{x}) - \nabla f(\boldsymbol{y})\| \leq L\|\boldsymbol{x} - \boldsymbol{y}\|. \tag{2}$$

A brief summary of related work appears in Section 6.

## 2 Motivating Real-world Examples

Many continuous objectives in practice turn out to be DR-submodular. Here we list several of them. More can be found in Appendix B.

**Softmax extension.** Determinantal point processes (DPPs) are probabilistic models of repulsion, that have been used to model diversity in machine learning [26]. The constrained MAP (maximum a posteriori) inference problem of a DPP is an NP-hard combinatorial problem in general. Currently, the methods with the best approximation guarantees are based on either maximizing the multilinear extension [6] or the softmax extension [20], both of which are DR-submodular functions (details in Appendix F.1). The multilinear extension is given as an expectation over the original set function values, thus evaluating the objective of this extension requires expensive sampling. In constast, the softmax extension has a closed form expression, which is much more appealing from a computational perspective. Let $\mathbf{L}$ be the positive semidefinite kernel matrix of a DPP, its softmax extension is:

$$f(\boldsymbol{x}) = \log \det \left( \operatorname{diag}(\boldsymbol{x})(\mathbf{L} - \mathbf{I}) + \mathbf{I} \right), \boldsymbol{x} \in [0,1]^n, \tag{3}$$

where $\mathbf{I}$ is the identity matrix, $\operatorname{diag}(\boldsymbol{x})$ is the diagonal matrix with diagonal elements set as $\boldsymbol{x}$. The problem of MAP inference in DPPs corresponds to the problem $\max_{\boldsymbol{x} \in \mathcal{P}} f(\boldsymbol{x})$, where $\mathcal{P}$ is a down-closed convex constraint, e.g., a matroid polytope or a matching polytope.

**Mean-field inference for log-submodular models.** Log-submodular models [9] are a class of probabilistic models over subsets of a ground set $\mathcal{V} = [n]$, where the log-densities are submodular set functions $F(S)$: $p(S) = \frac{1}{Z} \exp(F(S))$. The partition function $Z = \sum_{S \subseteq \mathcal{V}} \exp(F(S))$ is typically hard to evaluate. One can use mean-field inference to approximate $p(S)$ by some factorized distribution $q_{\boldsymbol{x}}(S) := \prod_{i \in S} x_i \prod_{j \notin S} (1 - x_j), \boldsymbol{x} \in [0,1]^n$, by minimizing the distance measured w.r.t. the Kullback-Leibler divergence between $q_{\boldsymbol{x}}$ and $p$, i.e., $\sum_{S \subseteq \mathcal{V}} q_{\boldsymbol{x}}(S) \log \frac{q_{\boldsymbol{x}}(S)}{p(S)}$. It is,

$$\mathrm{KL}(\boldsymbol{x}) = - \sum_{S \subseteq \mathcal{V}} \prod_{i \in S} x_i \prod_{j \notin S} (1 - x_j) F(S) + \sum_{i=1}^{n} [x_i \log x_i + (1 - x_i) \log(1 - x_i)] + \log Z.$$

$\mathrm{KL}(\boldsymbol{x})$ is DR-supermodular w.r.t. $\boldsymbol{x}$ (details in Appendix F.1). Minimizing the Kullback-Leibler divergence $\mathrm{KL}(\boldsymbol{x})$ amounts to maximizing a DR-submodular function.

### 2.1 Motivating Example Captured by Generalized Submodularity on Conic Lattices

Submodular continuous functions can already model many scenarios. Yet, there are several interesting cases which are in general not (DR-)Submodular, but can still be captured by a generalized notion. This generalization enables to develop polynomial algorithms with guarantees by using ideas from continuous submodular optimization. We present one representative objective here (more in Appendix B). In Appendix A we show the technical details on how they are covered by a class of submodular continuous functions over conic lattices.

Consider the logistic regression model with a *non-convex* separable regularizer. This flexibility may result in better statistical performance (e.g., in recovering discontinuities, [2]) compared to classical models with convex regularizers. Let $\boldsymbol{z}^1, ..., \boldsymbol{z}^m$ in $\mathbb{R}^n$ be $m$ training points with corresponding binary labels $\boldsymbol{y} \in \{\pm 1\}^m$. Assume that the following mild assumption is satisfied: For any fixed dimension $i$, all the data points have the same sign, i.e., $\operatorname{sign}(z_i^j)$ is the same for all $j \in [m]$ (which can be achieved by easily scaling if not). The task is to solve the following non-convex optimization problem,

$$\min_{\boldsymbol{x} \in \mathbb{R}^n} f(\boldsymbol{x}) := m^{-1} \sum_{j=1}^{m} f_j(\boldsymbol{x}) + \lambda r(\boldsymbol{x}), \tag{4}$$

where $f_j(\boldsymbol{x}) = \log(1 + \exp(-y_j \boldsymbol{x}^\top \boldsymbol{z}^j))$ is the logistic loss; $\lambda > 0$ is the regularization parameter, and $r(\boldsymbol{x})$ is some non-convex separable regularizer. Such separable regularizers are popular in

statistics, and two notable choices are $r(\boldsymbol{x}) = \sum_{i=1}^{n} \frac{\gamma x_i^2}{1 + \gamma x_i^2}$, and $r(\boldsymbol{x}) = \sum_{i=1}^{n} \min\{\gamma x_i^2, 1\}$ (see [2]). Let us define a vector $\boldsymbol{\alpha} \in \{\pm 1\}^n$ as $\alpha_i = \text{sign}(z_i^j), i \in [n]$ and $l(\boldsymbol{x}) := \frac{1}{m} \sum_{j=1}^{m} f_j(\boldsymbol{x})$. One can show that $l(\boldsymbol{x})$ is not DR-submodular or DR-supermodular. Yet, in Appendix A we will show that $l(\boldsymbol{x})$ is $\mathcal{K}_{\boldsymbol{\alpha}}$-DR-supermodular, where the latter generalizes DR-supermodularity. Usually, one can assume the optimal solution $\boldsymbol{x}^*$ lies in some box $[\underline{\boldsymbol{u}}, \bar{\boldsymbol{u}}]$. Then the problem is an instance of constrained non-monotone $\mathcal{K}_{\boldsymbol{\alpha}}$-DR-submodular maximization.

## 3 Underlying Properties of Constrained DR-submodular Maximization

In this section we present several properties arising in DR-submodular function maximization. First we show properties related to concavity of the objective along certain directions, then we establish the relation between locally stationary points and the global optimum (thus called "local-global relation"). These properties will be used to derive guarantees for the algorithms in Section 4. All omitted proofs are in Appendix D.

### 3.1 Properties Along Non-negative/Non-positive Directions

A DR-submodular function $f$ is *concave* along any non-negative/non-positive direction [4]. Notice that DR-submodularity is a stronger condition than concavity along directions $\boldsymbol{v} \in \pm\mathbb{R}_+^n$: for instance, a concave function is concave along any direction, but it may not be a DR-submodular function.

For a DR-submodular function with $L$-Lipschitz gradients, one can get the following quadratic lower bound using standard techniques by combing the concavity and Lipschitz gradients in (2).
**Quadratic lower bound.** If $f$ is DR-submodular with a $L$-Lipschitz gradient, then for all $\boldsymbol{x} \in \mathcal{X}$ and $\boldsymbol{v} \in \pm\mathbb{R}_+^n$, it holds,

$$f(\boldsymbol{x} + \boldsymbol{v}) \geq f(\boldsymbol{x}) + \langle \nabla f(\boldsymbol{x}), \boldsymbol{v} \rangle - \frac{L}{2} \|\boldsymbol{v}\|^2. \tag{5}$$

It will be used in Section 4.2 for analyzing the non-monotone FRANK-WOLFE variant (Algorithm 2).

**Strong DR-submodularity.** DR-submodular objectives may be strongly concave along directions $\boldsymbol{v} \in \pm\mathbb{R}_+^n$, e.g., for DR-submodular quadratic functions. We will show that such additional structure may be exploited to obtain stronger guarantees for the local-global relation.

**Definition 3** (Strongly DR-submodular). *A function $f$ is $\mu$-strongly DR-submodular ($\mu \geq 0$) if for all $\boldsymbol{x} \in \mathcal{X}$ and $\boldsymbol{v} \in \pm\mathbb{R}_+^n$, it holds that,*

$$f(\boldsymbol{x} + \boldsymbol{v}) \leq f(\boldsymbol{x}) + \langle \nabla f(\boldsymbol{x}), \boldsymbol{v} \rangle - \frac{\mu}{2} \|\boldsymbol{v}\|^2. \tag{6}$$

### 3.2 Relation Between Approximately Stationary Points and Global Optimum

First of all, we present the following Lemma, which will motivate us to consider a non-stationarity measure for general constrained optimization problems.

**Lemma 1.** *If $f$ is $\mu$-strongly DR-submodular, then for any two points $\boldsymbol{x}$, $\boldsymbol{y}$ in $\mathcal{X}$, it holds:*

$$(\boldsymbol{y} - \boldsymbol{x})^\top \nabla f(\boldsymbol{x}) \geq f(\boldsymbol{x} \vee \boldsymbol{y}) + f(\boldsymbol{x} \wedge \boldsymbol{y}) - 2f(\boldsymbol{x}) + \frac{\mu}{2} \|\boldsymbol{x} - \boldsymbol{y}\|^2. \tag{7}$$

Lemma 1 implies that if $\boldsymbol{x}$ is stationary (i.e., $\nabla f(\boldsymbol{x}) = 0$), then $2f(\boldsymbol{x}) \geq f(\boldsymbol{x} \vee \boldsymbol{y}) + f(\boldsymbol{x} \wedge \boldsymbol{y}) + \frac{\mu}{2} \|\boldsymbol{x} - \boldsymbol{y}\|^2$, which gives an implicit relation between $\boldsymbol{x}$ and $\boldsymbol{y}$. While in practice finding an exact stationary point is not easy, usually non-convex solvers will arrive at an approximately stationary point, thus requiring a proper measure of non-stationarity for the constrained optimization problem.

**Non-stationarity measure.** Looking at the LHS of (7), it naturally suggests to use $\max_{\boldsymbol{y} \in \mathcal{P}} (\boldsymbol{y} - \boldsymbol{x})^\top \nabla f(\boldsymbol{x})$ as the non-stationarity measure, which happens to coincide with the measure proposed by recent work of [27], and it can be calculated for free for Frank-Wolfe-style algorithms (e.g., Algorithm 3). In order to adapt it to the local-global relation, we give a slightly more general definition here: For any constraint set $\mathcal{Q} \subseteq \mathcal{X}$, the non-stationarity of a point $\boldsymbol{x} \in \mathcal{Q}$ is,

$$g_{\mathcal{Q}}(\boldsymbol{x}) := \max_{\boldsymbol{v} \in \mathcal{Q}} \langle \boldsymbol{v} - \boldsymbol{x}, \nabla f(\boldsymbol{x}) \rangle \qquad \text{(non-stationarity).} \tag{8}$$

It always holds that $g_{\mathcal{Q}}(\boldsymbol{x}) \geq 0$, and $\boldsymbol{x}$ is a stationary point in $\mathcal{Q}$ iff $g_{\mathcal{Q}}(\boldsymbol{x}) = 0$, so (8) is a natural generalization of the non-stationarity measure $\|\nabla f(\boldsymbol{x})\|$ for unconstrained optimization. As the next statement shows, $g_{\mathcal{Q}}(\boldsymbol{x})$ plays an important role in characterizing the local-global relation.

**Proposition 1** (Local-Global Relation). *Let $\boldsymbol{x}$ be a point in $\mathcal{P}$ with non-stationarity $g_{\mathcal{P}}(\boldsymbol{x})$, and $\mathcal{Q} := \{\boldsymbol{y} \in \mathcal{P} \mid \boldsymbol{y} \leq \bar{\boldsymbol{u}} - \boldsymbol{x}\}$. Let $\boldsymbol{z}$ be a point in $\mathcal{Q}$ with non-stationarity $g_{\mathcal{Q}}(\boldsymbol{z})$. It holds that,*

$$\max\{f(\boldsymbol{x}), f(\boldsymbol{z})\} \geq \frac{1}{4}\left[f(\boldsymbol{x}^*) - g_{\mathcal{P}}(\boldsymbol{x}) - g_{\mathcal{Q}}(\boldsymbol{z})\right] + \frac{\mu}{8}\left(\|\boldsymbol{x} - \boldsymbol{x}^*\|^2 + \|\boldsymbol{z} - \boldsymbol{z}^*\|^2\right), \quad (9)$$

*where $\boldsymbol{z}^* := \boldsymbol{x} \vee \boldsymbol{x}^* - \boldsymbol{x}$.*

**Proof sketch of Proposition 1:** The proof uses Lemma 1, the non-stationarity in (8) and a key observation in the following Claim. The detailed proof is in Appendix D.2.

**Claim 1.** *It holds that $f(\boldsymbol{x} \vee \boldsymbol{x}^*) + f(\boldsymbol{x} \wedge \boldsymbol{x}^*) + f(\boldsymbol{z} \vee \boldsymbol{z}^*) + f(\boldsymbol{z} \wedge \boldsymbol{z}^*) \geq f(\boldsymbol{x}^*)$.*

Note that [7; 20] propose a similar relation for the special cases of multilinear/softmax extensions by mainly proving the same conclusion as in Claim 1. Their relation does not incorporate the properties of non-stationarity or strong DR-submodularity. They both use the proof idea of constructing a complicated auxiliary set function tailored to specific DR-submodular functions. We present a different proof method by directly utilizing the DR property on carefully constructed auxiliary points (e.g., $(\boldsymbol{x} + \boldsymbol{z}) \vee \boldsymbol{x}^*$ in the proof of Claim 1).

# 4 Algorithms for Constrained DR-submodular Maximization

Based on the properties, we present two algorithms for solving (P). The first is based on the local-global relation, and the second is a FRANK-WOLFE variant adapted for the non-monotone setting. All the omitted proofs are deferred to Appendix E.

## 4.1 An Algorithm Based on the Local-Global Relation

---

**Algorithm 1:** TWO-PHASE FRANK-WOLFE for non-monotone DR-submodular maximization

---

**Input**: $\max_{\boldsymbol{x} \in \mathcal{P}} f(\boldsymbol{x})$, stopping tolerance $\epsilon_1, \epsilon_2$, #iterations $K_1, K_2$

1  $\boldsymbol{x} \leftarrow$ NON-CONVEX FRANK-WOLFE$(f, \mathcal{P}, K_1, \epsilon_1, \boldsymbol{x}^{(0)})$ ;       // $\boldsymbol{x}^{(0)} \in \mathcal{P}$
2  $\mathcal{Q} \leftarrow \mathcal{P} \cap \{\boldsymbol{y} \in \mathbb{R}_+^n \mid \boldsymbol{y} \leq \bar{\boldsymbol{u}} - \boldsymbol{x}\}$;
3  $\boldsymbol{z} \leftarrow$ NON-CONVEX FRANK-WOLFE$(f, \mathcal{Q}, K_2, \epsilon_2, \boldsymbol{z}^{(0)})$ ;       // $\boldsymbol{z}^{(0)} \in \mathcal{Q}$

**Output**: $\arg\max\{f(\boldsymbol{x}), f(\boldsymbol{z})\}$ ;

---

We summarize the TWO-PHASE algorithm in Algorithm 1. It is generalized from the "two-phase" method in [7; 20]. It invokes some non-convex solver (we use the NON-CONVEX FRANK-WOLFE by [27]; pseudocode is included in Algorithm 3 of Appendix C) to find approximately stationary points in $\mathcal{P}$ and $\mathcal{Q}$, respectively, then returns the solution with the larger function value. Though we use NON-CONVEX FRANK-WOLFE as the subroutine here, it is worth noting that any algorithm that is guaranteed to find an approximately stationary point can be plugged into Algorithm 1 as the subroutine. We give an improved approximation bound by considering more properties of DR-submodular functions. Borrowing the results from [27] for the NON-CONVEX FRANK-WOLFE subroutine, we get the following,

**Theorem 1.** *The output of Algorithm 1 satisfies,*

$$\max\{f(\boldsymbol{x}), f(\boldsymbol{z})\} \geq \frac{\mu}{8}\left(\|\boldsymbol{x} - \boldsymbol{x}^*\|^2 + \|\boldsymbol{z} - \boldsymbol{z}^*\|^2\right) \quad (10)$$

$$+ \frac{1}{4}\left[f(\boldsymbol{x}^*) - \min\left\{\frac{\max\{2h_1, C_f(\mathcal{P})\}}{\sqrt{K_1 + 1}}, \epsilon_1\right\} - \min\left\{\frac{\max\{2h_2, C_f(\mathcal{Q})\}}{\sqrt{K_2 + 1}}, \epsilon_2\right\}\right],$$

*where $h_1 := \max_{\boldsymbol{x} \in \mathcal{P}} f(\boldsymbol{x}) - f(\boldsymbol{x}^{(0)})$, $h_2 := \max_{\boldsymbol{z} \in \mathcal{Q}} f(\boldsymbol{z}) - f(\boldsymbol{z}^{(0)})$ are the initial suboptimalities, $C_f(\mathcal{P}) := \sup_{\boldsymbol{x}, \boldsymbol{v} \in \mathcal{P}, \gamma \in [0,1], \boldsymbol{y} = \boldsymbol{x} + \gamma(\boldsymbol{v} - \boldsymbol{x})} \frac{2}{\gamma^2}(f(\boldsymbol{y}) - f(\boldsymbol{x}) - (\boldsymbol{y} - \boldsymbol{x})^\top \nabla f(\boldsymbol{x}))$ is the curvature of $f$ w.r.t. $\mathcal{P}$, and $\boldsymbol{z}^* = \boldsymbol{x} \vee \boldsymbol{x}^* - \boldsymbol{x}$.*

Theorem 1 indicates that Algorithm 1 has a 1/4 approximation guarantee and $1/\sqrt{k}$ rate. However, it has good empirical performance as demonstrated by the experiments in Section 5. Informally, this can be partially explained by the term $\frac{\mu}{8}\left(\|x - x^*\|^2 + \|z - z^*\|^2\right)$ in (10): if $x$ is away from $x^*$, this term will augment the bound; if $x$ is close to $x^*$, by the smoothness of $f$, it should be close to optimal.

## 4.2 The Non-monotone FRANK-WOLFE Variant

---

**Algorithm 2:** Non-monotone FRANK-WOLFE variant for DR-submodular maximization

**Input**: $\max_{x\in\mathcal{P}} f(x)$, prespecified step size $\gamma \in (0, 1]$

1  $x^{(0)} \leftarrow 0$, $t^{(0)} \leftarrow 0$, $k \leftarrow 0$;                              // $k$ : iteration index, $t^{(k)}$ : cumulative step size

2  **while** $t^{(k)} < 1$ **do**

3  $\quad$ $v^{(k)} \leftarrow \arg\max_{v\in\mathcal{P}, v\leq\bar{u}-x^{(k)}}\langle v, \nabla f(x^{(k)})\rangle$;                              // shrunken LMO

4  $\quad$ use uniform step size $\gamma_k = \gamma$; set $\gamma_k \leftarrow \min\{\gamma_k, 1 - t^{(k)}\}$;

5  $\quad$ $x^{(k+1)} \leftarrow x^{(k)} + \gamma_k v^{(k)}$, $t^{(k+1)} \leftarrow t^{(k)} + \gamma_k$, $k \leftarrow k + 1$;

**Output**: $x^{(K)}$ ;                              // assuming there are $K$ iterations in total

---

Algorithm 2 summarizes the non-monotone FRANK-WOLFE variant, which is inspired by the unified continuous greedy algorithm in [13] for maximizing the multilinear extension of a submodular set function. It initializes the solution $x^{(0)}$ to be 0, and maintains $t^{(k)}$ as the cumulative step size. At iteration $k$, it maximizes the linearization of $f$ over a "shrunken" constraint set: $\{v|v \in \mathcal{P}, v \leq \bar{u} - x^{(k)}\}$, which is different from the classical LMO of Frank-Wolfe-style algorithms (hence we refer to it as the "shrunken LMO"). Then it employs an update step in the direction $v^{(k)}$ chosen by the LMO with a uniform step size $\gamma_k = \gamma$. The cumulative step size $t^{(k)}$ is used to ensure that the overall step sizes sum to one, thus the output solution $x^{(K)}$ is a convex combination of the LMO outputs, hence also lies in $\mathcal{P}$.

The shrunken LMO (Step 3) is the key difference compared to the monotone FRANK-WOLFE variant in [4]. The extra constraint $v \leq \bar{u} - x^{(k)}$ is added to prevent too aggressive growth of the solution, since in the non-monotone setting such aggressive growth may hurt the overall performance. The next theorem states the guarantees of Algorithm 2.

**Theorem 2.** *Consider Algorithm 2 with uniform step size $\gamma$. For $k = 1, ..., K$ it holds that,*

$$f(x^{(k)}) \geq t^{(k)}e^{-t^{(k)}}f(x^*) - \frac{LD^2}{2}k\gamma^2 - O(\gamma^2)f(x^*). \tag{11}$$

By observing that $t^{(K)} = 1$ and applying Theorem 2, we get the following Corollary:

**Corollary 1.** *The output of Algorithm 2 satisfies $f(x^{(K)}) \geq e^{-1}f(x^*) - \frac{LD^2}{2K} - O\left(\frac{1}{K^2}\right)f(x^*)$.*

Corollary 1 shows that Algorithm 2 enjoys a sublinear convergence rate towards some point $x^{(K)}$ inside $\mathcal{P}$, with a $1/e$ approximation guarantee.

**Proof sketch of Theorem 2:** The proof is by induction. To prepare the building blocks, we first of all show that the growth of $x^{(k)}$ is indeed bounded,

**Lemma 2.** *Assume $x^{(0)} = 0$. For $k = 0, ..., K - 1$, it holds $x_i^{(k)} \leq \bar{u}_i[1 - (1 - \gamma)^{t^{(k)}/\gamma}], \forall i \in [n]$.*

Then the following Lemma provides a lower bound, which gets the global optimum involved,

**Lemma 3** (Generalized from Lemma 7 in [8])**.** *Given $\theta \in (0, \bar{u}]$, let $\lambda' = \min_{i\in[n]} \frac{\bar{u}_i}{\theta_i}$. Then for all $x \in [0, \theta]$, it holds $f(x \vee x^*) \geq (1 - \frac{1}{\lambda'})f(x^*)$.*

Then the key ingredient for induction is the relation between $f(x^{(k+1)})$ and $f(x^{(k)})$ indicated by:

**Claim 2.** *For $k = 0, ..., K-1$ it holds $f(x^{(k+1)}) \geq (1-\gamma)f(x^{(k)}) + \gamma(1-\gamma)^{t^{(k)}/\gamma}f(x^*) - \frac{LD^2}{2}\gamma^2$,*

which is derived by a combination of the quadratic lower bound in (5), Lemma 2 and Lemma 3.

**Remarks on the two algorithms.** Notice that though the TWO-PHASE algorithm has a worse guarantee than the non-monotone FRANK-WOLFE variant, it is still of interest: i) It allows flexibility in using a wide range of existing solvers for finding an (approximately) stationary point. ii) The guarantees that we present rely on a worst-case analysis. The empirical performance of the TWO-PHASE algorithm is often comparable or better than that of the FRANK-WOLFE variant. This suggests to explore more properties in concrete problems that may favor the TWO-PHASE algorithm, which we leave for future work.

## 5  Experimental Results

We test the performance of the analyzed algorithms, while considering the following baselines: 1) QUADPROGIP [39], which is a global solver for non-convex quadratic programming; 2) Projected gradient ascent (PROJGRAD) with diminishing step sizes ($\frac{1}{k+1}$, $k$ starts from 0). We run all the algorithms for 100 iterations. For the subroutine (Algorithm 3) of TWO-PHASE FRANK-WOLFE, we set $\epsilon_1 = \epsilon_2 = 10^{-6}, K_1 = K_2 = 100$. All the synthetic results are the average of 20 repeated experiments. All experiments were implemented using MATLAB. Source code can be found at:
https://github.com/bianan/non-monotone-dr-submodular.

### 5.1  DR-submodular Quadratic Programming

As a state-of-the-art global solver, QUADPROGIP[2] [39] can find the global optimum (possibly in exponential time), which were used to calculate the approximation ratios. Our problem instances are synthetic DR-submodular quadratic objectives with down-closed polytope constraints, i.e., $f(\boldsymbol{x}) = \frac{1}{2}\boldsymbol{x}^\top \mathbf{H}\boldsymbol{x} + \boldsymbol{h}^\top \boldsymbol{x} + c$ and $\mathcal{P} = \{\boldsymbol{x} \in \mathbb{R}^n_+ \mid \mathbf{A}\boldsymbol{x} \leq \boldsymbol{b}, \boldsymbol{x} \leq \bar{\boldsymbol{u}}, \mathbf{A} \in \mathbb{R}^{m \times n}_{++}, \boldsymbol{b} \in \mathbb{R}^m_+\}$. Both objective and constraints were randomly generated, in the following two manners:

**1) Uniform distribution.** $\mathbf{H} \in \mathbb{R}^{n \times n}$ is a symmetric matrix with uniformly distributed entries in $[-1, 0]$; $\mathbf{A} \in \mathbb{R}^{m \times n}$ has uniformly distributed entries in $[\nu, \nu + 1]$, where $\nu = 0.01$ is a small positive constant in order to make entries of $\mathbf{A}$ strictly positive.

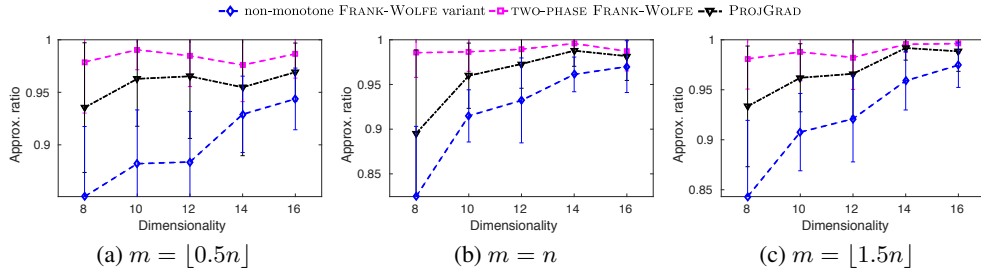

Figure 1: Results on DR-submodular quadratic instances with uniform distribution.

**2) Exponential distribution.** The entries of $-\mathbf{H}$ and $\mathbf{A}$ were sampled from exponential distributions Exp($\lambda$) (For a random variable $y \geq 0$, its probability density function is $\lambda e^{-\lambda y}$, and for $y < 0$, its density is 0). Specifically, each entry of $-\mathbf{H}$ was sampled from Exp(1), then the matrix $-\mathbf{H}$ was made to be symmetric. Each entry of $\mathbf{A}$ was sampled from Exp(0.25) $+ \nu$, where $\nu = 0.01$ is a small positive constant.

In both the above two cases, we set $\boldsymbol{b} = \mathbf{1}^m$, and $\bar{\boldsymbol{u}}$ to be the tightest upper bound of $\mathcal{P}$ by $\bar{u}_j = \min_{i \in [m]} \frac{b_i}{A_{ij}}, \forall j \in [n]$. In order to make $f$ non-monotone, we set $\boldsymbol{h} = -0.2 * \mathbf{H}^\top \bar{\boldsymbol{u}}$. To make sure that $f$ is non-negative, we first of all solve the problem $\min_{\boldsymbol{x} \in \mathcal{P}} \frac{1}{2}\boldsymbol{x}^\top \mathbf{H}\boldsymbol{x} + \boldsymbol{h}^\top \boldsymbol{x}$ using QUADPROGIP, let the solution to be $\hat{\boldsymbol{x}}$, then set $c = -f(\hat{\boldsymbol{x}}) + 0.1 * |f(\hat{\boldsymbol{x}})|$.

The approximation ratios w.r.t. dimensionalities ($n$) are plotted in Figures 1 and 2, for the two manners of data generation. We set the number of constraints to be $m = \lfloor 0.5n \rfloor$, $m = n$ and $m = \lfloor 1.5n \rfloor$ in Figures 1a to 1c (and Figures 2a to 2c), respectively.

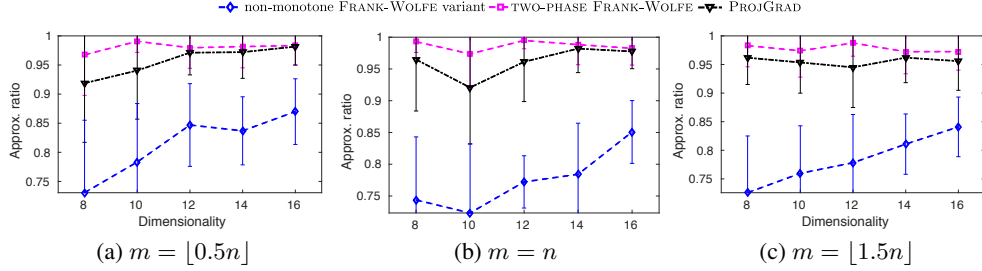

Figure 2: Results on quadratic instances with exponential distribution.

One can see that TWO-PHASE FRANK-WOLFE usually performs the best, PROJGRAD follows, and non-monotone FRANK-WOLFE variant is the last. The good performance of TWO-PHASE FRANK-WOLFE can be partially explained by the strong DR-submodularity of quadratic functions according to Theorem 1. Performance of the two analyzed algorithms is consistent with the theoretical bounds: the approximation ratios of FRANK-WOLFE variant are always much higher than $1/e$.

## 5.2 Maximizing Softmax Extensions

With some derivation, one can see the derivative of the softmax extension in (3) is: $\nabla_i f(\boldsymbol{x}) = \text{tr}((\text{diag}(\boldsymbol{x})(\mathbf{L} - \mathbf{I}) + \mathbf{I})^{-1}(\mathbf{L} - \mathbf{I})_i), \forall i \in [n]$, where $(\mathbf{L} - \mathbf{I})_i$ denotes the matrix obtained by zeroing all entries except the $i$th row of $(\mathbf{L} - \mathbf{I})$. Let $\mathbf{C} := (\text{diag}(\boldsymbol{x})(\mathbf{L} - \mathbf{I}) + \mathbf{I})^{-1}, \mathbf{D} := (\mathbf{L} - \mathbf{I})$, one can see that $\nabla_i f(\boldsymbol{x}) = \mathbf{D}_{i \cdot}^{\top} \mathbf{C}_{\cdot i}$, which gives an efficient way to calculate the gradient $\nabla f(\boldsymbol{x})$.

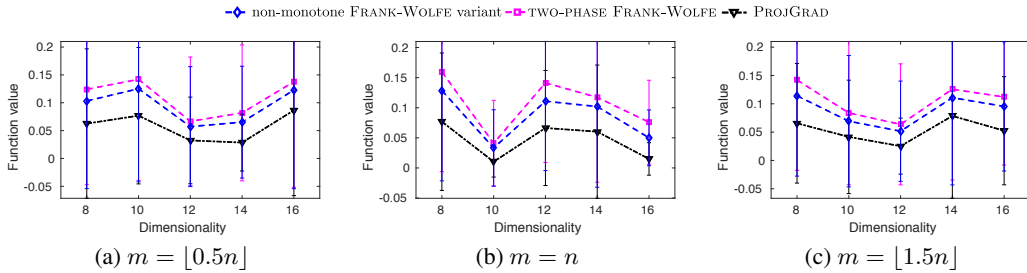

Figure 3: Results on softmax instances with polytope constraints generated from uniform distribution.

**Results on synthetic data.** We generate the softmax objectives (see (3)) in the following way: first generate the $n$ eigenvalues $\boldsymbol{d} \in \mathbb{R}_+^n$, each randomly distributed in $[0, 1.5]$, and set $\mathbf{D} = \text{diag}(\boldsymbol{d})$. After generating a random unitary matrix $\mathbf{U}$, we set $\mathbf{L} = \mathbf{U}\mathbf{D}\mathbf{U}^{\top}$. One can verify that $\mathbf{L}$ is positive semidefinite and has eigenvalues as the entries of $\boldsymbol{d}$.

We generate the down-closed polytope constraints in the same form and same way as that for DR-submodular quadratic functions, except for setting $\boldsymbol{b} = 2 * \mathbf{1}^m$. Function values returned by different solvers w.r.t. $n$ are shown in Figure 3, for which the random polytope constraints were generated with uniform distribution (results for which the random polytope constraints were generated with exponential distribution are deferred to Appendix G). The number of constraints was set to be $m = \lfloor 0.5n \rfloor$, $m = n$ and $m = \lfloor 1.5n \rfloor$ in Figures 3a to 3c, respectively. One can observe that TWO-PHASE FRANK-WOLFE still has the best performance, the non-monotone FRANK-WOLFE variant follows, and PROJGRAD has the worst performance.

**Real-world results on matched summarization.** The task of "matched summarization" is to select a set of document *pairs* out of a corpus of documents, such that the two documents within a pair are similar, and the overall set of pairs is as diverse as possible. The motivation for this task is very practical: it could be, for example, to compare the opinions of various politicians on a range of representative topics.

In our experiments, we used a similar setting to the one in [20]. We experimented on the 2012 US Republican debates data, which consists of 8 candidates: Bachman, Gingrich, Huntsman, Paul, Perry, Romney and Santorum. Each task involves one pair of candidates, so in total there are

$28 = 7 * 8/2$ tasks. Figure 4a plots the averaged function values returned by the three solvers over 28 tasks, w.r.t. different values of a hyperparameter reflecting the matching quality (details see [20]).

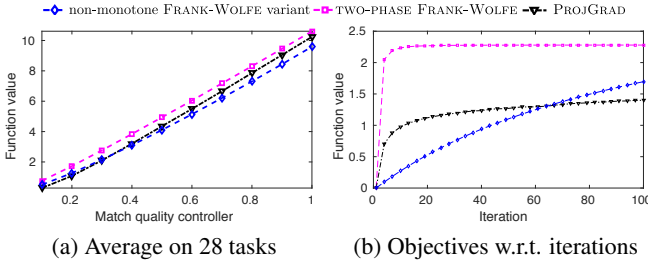

(a) Average on 28 tasks   (b) Objectives w.r.t. iterations

Figure 4: Results on 2012 US Republican debates data.

Figure 4b traces the objectives w.r.t. iterations for a specific candidate pair (Bachman, Romney). For TWO-PHASE FRANK-WOLFE, the objectives of the selected phase were plotted. One can see that TWO-PHASE FRANK-WOLFE also achieves the best performance, while the performance of non-monotone FRANK-WOLFE variant and PROJGRAD is comparable.

## 6 Related Work

Submodular optimization and, more broadly, non-convex optimization are extensively studied in the literature, which renders it very difficult comprehensively surveying all previous work. Here we only briefly summarize some of the most related papers.

**Submodular optimization over integer-lattice and continuous domains.** Many results from submodular set function optimization have been generalized to the integer-lattice case [34; 33; 12; 24]. Of particular interest is the reduction [12] from an integer-lattice DR-submodular maximization problem to a submodular set function maximization problem. Submodular optimization over continuous domains has attracted considerable attention recently [3; 4; 36]. Two classes of functions that are covered by continuous submodularity are the Lovasz extensions [28] and multilinear extensions [6] of submodular set functions. Particularly, multilinear extensions of submodular set functions are also continuous DR-submodular [3], but with the special property that they are coordinate-wise linear. Combined with the rounding technique of contention resolution [7], maximizing multilinear extensions [38; 19; 13; 8; 11] has become the state-of-the-art method for submodular set function maximization. Some of the techniques in maximizing multilinear extensions [13; 7; 8] have inspired this work. However, we are the first to explore the rich properties and devise algorithms for the general constrained DR-submodular maximization problem over continuous domains.

**Non-convex optimization.** Non-convex optimization receives a surge of attention in the past years. One active research topic is to reach a stationary point for unconstrained optimization [35; 32; 1] or constrained optimization [18; 27]. However, without proper assumptions, a stationary point may not lead to any global approximation guarantee. The local-global relation (in Proposition 1) provides a strong relation between (approximately) stationary points and global optimum, thus making it flexible to incorporate progress in this area.

## 7 Conclusion

We have studied the problem of constrained non-monotone DR-submodular continuous maximization. We explored the structural properties of such problems, and established a local-global relation. Based on these properties, we presented a TWO-PHASE algorithm with a $1/4$ approximation guarantee, and a non-monotone FRANK-WOLFE variant with a $1/e$ approximation guarantee. We further generalized submodular continuous function over conic lattices, which enabled us to model a larger class of applications. Lastly, our theoretical findings were verified by synthetic and real-world experiments.

**Acknowledgement.** This research was partially supported by ERC StG 307036, by the Max Planck ETH Center for Learning Systems, and by the ETH Zürich Postdoctoral Fellowship program.

## Footnotes

[1]Since otherwise one can work on a new function $g(\boldsymbol{x}) := f(\boldsymbol{x} + \underline{\boldsymbol{u}})$ that has 0 as the lower bound of its domain, and all properties of the function are still preserved.

[2]We used the open source code provided by [39], and the IBM CPLEX optimization studio https://www.ibm.com/jm-en/marketplace/ibm-ilog-cplex as the subroutine.

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
