[Supplementary Material]

# Appendix

## A  Submodular Continuous Functions on Conic Lattices and the Reduction

Motivated by the objectives that can not be modeled by continuous submodular functions, we consider the more general submodular continuous functions over lattices induced by conic inequalities. Furthermore we provide a reduction to the original (DR-)submodular optimization problem.

### A.1  Definitions and Properties

Let us look at the proper cone that will be used to define a conic inequality firstly. A cone $\mathcal{K} \subseteq \mathbb{R}^n$ is a *proper cone* if it is convex, closed, solid (having nonempty interior) and pointed (contains no line, i.e., $\boldsymbol{x} \in \mathcal{K}, -\boldsymbol{x} \in \mathcal{K}$ implies $\boldsymbol{x} = 0$). A proper cone $\mathcal{K}$ can be used to define a conic inequality (a.k.a. generalized inequality [5, Chapter 2.4]): $\boldsymbol{a} \preceq_{\mathcal{K}} \boldsymbol{b}$ iff $\boldsymbol{b} - \boldsymbol{a} \in \mathcal{K}$, which also defines a partial ordering since the binary relation $\preceq_{\mathcal{K}}$ is reflexive, antisymmetric and transitive. Then it is easy to see that $(\mathcal{X}, \preceq_{\mathcal{K}})$ is a partial ordered set (poset).

If two elements $\boldsymbol{a}, \boldsymbol{b} \in \mathcal{X}$ have a least upper bound (greatest lower bound), it is denoted as the "join": $\boldsymbol{a} \vee \boldsymbol{b}$ (the "meet": $\boldsymbol{a} \wedge \boldsymbol{b}$). A lattice is a poset that contains the join and meet of each pair of its elements [17]. A "lattice cone" [15] is the proper cone that can be used to define a lattice. Note that not all conic inequalities can be used to define a lattice. For example, the positive semidefine cone $\mathcal{K}_{\mathrm{PSD}} = \{\mathbf{A} \in \mathbb{R}^{n \times n} | \mathbf{A} \text{ is symmetric}, \mathbf{A} \succeq 0\}$ is a proper cone, but its induced ordering can not be used to define a lattice. There is a simple counter example to show this in Appendix F.2.

Specifically, we name the lattice that can be defined through a conic inequality as "conic lattice", since it is of particular interest for modeling the real-world applications in this paper.

**Definition 4** (Conic Lattice). *Given a poset $(\mathcal{X}, \preceq_{\mathcal{K}})$ induced by the conic inequality $\preceq_{\mathcal{K}}$, if there exist joint and meet operations for every pair of elements $(\boldsymbol{a}, \boldsymbol{b})$ in $\mathcal{X} \times \mathcal{X}$, s.t. $\boldsymbol{a} \vee \boldsymbol{b}$ and $\boldsymbol{a} \wedge \boldsymbol{b}$ are still in $\mathcal{X}$, then we call $(\mathcal{X}, \preceq_{\mathcal{K}})$ a conic lattice.*

In one word, a conic lattice $(\mathcal{X}, \preceq_{\mathcal{K}})$ is a lattice induced by a conic inequality $\preceq_{\mathcal{K}}$. In the following we introduce a class of conic lattices to model the applications in this work. We further provide a general characterization about submodularity on this conic lattice.

**Orthant conic lattice.** Given a sign vector $\boldsymbol{\alpha} \in \{\pm 1\}^n$, the orthant cone is defined as $\mathcal{K}_{\boldsymbol{\alpha}} := \{\boldsymbol{x} \in \mathbb{R}^n \mid x_i \alpha_i \geq 0, \forall i \in [n]\}$, one can see that $\mathcal{K}_{\boldsymbol{\alpha}}$ is a proper cone. For any two points $\boldsymbol{a}, \boldsymbol{b} \in \mathcal{X}$, one can further define the join and meet operations: $(\boldsymbol{a} \vee \boldsymbol{b})_i := \alpha_i \max\{\alpha_i a_i, \alpha_i b_i\}$, $(\boldsymbol{a} \wedge \boldsymbol{b})_i := \alpha_i \min\{\alpha_i a_i, \alpha_i b_i\}, \forall i \in [n]$. Then one can show that the poset $(\mathcal{X}, \preceq_{\mathcal{K}_{\boldsymbol{\alpha}}})$ is a conic lattice.

A function $f : \mathcal{X} \mapsto \mathbb{R}$ is submodular on a lattice [37; 16] if for all $(\boldsymbol{x}, \boldsymbol{y}) \in \mathcal{X} \times \mathcal{X}$, it holds that,

$$f(\boldsymbol{x}) + f(\boldsymbol{y}) \geq f(\boldsymbol{x} \vee \boldsymbol{y}) + f(\boldsymbol{x} \wedge \boldsymbol{y}). \tag{12}$$

One can establish the characterizations of submodularity on the orthant conic lattice $(\mathcal{X}, \preceq_{\mathcal{K}_{\boldsymbol{\alpha}}})$ similarly as that in [4]:

**Proposition 2** (Characterizations of Submodularity on Orthant Conic Lattice $(\mathcal{X}, \preceq_{\mathcal{K}_{\boldsymbol{\alpha}}})$). *If a function $f$ is is submodular on the lattice $(\mathcal{X}, \preceq_{\mathcal{K}_{\boldsymbol{\alpha}}})$ (called $\mathcal{K}_{\boldsymbol{\alpha}}$-submodular), then we have the following two equivalent characterizations:*
*a) $\forall \boldsymbol{a}, \boldsymbol{b} \in \mathcal{X}$ s.t. $\boldsymbol{a} \preceq_{\mathcal{K}_{\boldsymbol{\alpha}}} \boldsymbol{b}, \forall i$ s.t. $a_i = b_i, \forall k \in \mathbb{R}_+$ s.t. $(k\boldsymbol{e}_i + \boldsymbol{a})$ and $(k\boldsymbol{e}_i + \boldsymbol{b})$ are still in $\mathcal{X}$, it holds that,*

$$\alpha_i[f(k\boldsymbol{e}_i + \boldsymbol{a}) - f(\boldsymbol{a})] \geq \alpha_i[f(k\boldsymbol{e}_i + \boldsymbol{b}) - f(\boldsymbol{b})]. \quad \text{(weak DR)} \tag{13}$$

*b) If $f$ is twice differentiable, then $\forall \boldsymbol{x} \in \mathcal{X}$ it holds,*

$$\alpha_i \alpha_j \nabla^2_{ij} f(\boldsymbol{x}) \leq 0, \ \forall i, j \in [n], i \neq j. \tag{14}$$

Proposition 2 can be proved by directly generalizing the proof of Proposition 1 in [4], proof is omitted here due to the high similarity. Next, we generalize the definition of DR-submodularity to the conic lattice $(\mathcal{X}, \preceq_{\mathcal{K}_{\boldsymbol{\alpha}}})$:

**Definition 5** ($\mathcal{K}_{\boldsymbol{\alpha}}$-DR-submodular). *A function $f : \mathcal{X} \mapsto \mathbb{R}$ is $\mathcal{K}_{\boldsymbol{\alpha}}$-DR-submodular if $\forall \boldsymbol{a}, \boldsymbol{b} \in \mathcal{X}$ s.t. $\boldsymbol{a} \preceq_{\mathcal{K}_{\boldsymbol{\alpha}}} \boldsymbol{b}$, $\forall i \in [n], \forall k \in \mathbb{R}_+$ s.t. $(k\boldsymbol{e}_i + \boldsymbol{a})$ and $(k\boldsymbol{e}_i + \boldsymbol{b})$ are still in $\mathcal{X}$, it holds that,*

$$\alpha_i[f(k\boldsymbol{e}_i + \boldsymbol{a}) - f(\boldsymbol{a})] \geq \alpha_i[f(k\boldsymbol{e}_i + \boldsymbol{b}) - f(\boldsymbol{b})]. \tag{15}$$

In correspondence to the relation between DR-submodularity and submodularity over continuous domains (Proposition 2 in [4]), one can easily get the similar relation (with highly similar proof) in bellow:

**Proposition 3** ($\mathcal{K}_{\boldsymbol{\alpha}}$-submodular + coordinate-wise concave $\Leftrightarrow$ $\mathcal{K}_{\boldsymbol{\alpha}}$-DR-submodular). *A function $f$ is $\mathcal{K}_{\boldsymbol{\alpha}}$-DR-submodular iff it is $\mathcal{K}_{\boldsymbol{\alpha}}$-submodular and coordinate-wise concave.*

Combining (14) and Proposition 3, one can show that if $f$ is twice differentiable and $\mathcal{K}_{\boldsymbol{\alpha}}$-DR-submodular, then $\forall \boldsymbol{x} \in \mathcal{X}$ it holds that,

$$\alpha_i \alpha_j \nabla_{ij}^2 f(\boldsymbol{x}) \leq 0, \ \forall i, j \in [n]. \tag{16}$$

Similarly, a function $f$ is $\mathcal{K}_{\boldsymbol{\alpha}}$-DR-supermodular iff $-f$ is $\mathcal{K}_{\boldsymbol{\alpha}}$-DR-submodular.

**Remark:** We only consider the orthant conic lattice $(\mathcal{X}, \preceq_{\mathcal{K}_{\boldsymbol{\alpha}}})$ here, since it can already model the applications in this paper. However, it is noteworthy that the framework can be generalized to arbitrary conic lattices, which may be of interest to model more complicated applications. We left this as future exploration.

## A.2 A Reduction to Optimizing Submodular Functions over Continuous Domains

To be succint, in this section we only discuss the reduction for the $\mathcal{K}_{\boldsymbol{\alpha}}$-DR-submodular maximization problems. However, it is easy to see that the reduction works for all kinds of $\mathcal{K}_{\boldsymbol{\alpha}}$-submodular optimization problems, e.g., $\mathcal{K}_{\boldsymbol{\alpha}}$-submodular minimization problem.

Suppose $g$ is a $\mathcal{K}_{\boldsymbol{\alpha}}$-DR-submodular function, and the $\mathcal{K}_{\boldsymbol{\alpha}}$-DR-submodular maximization problem is $\max_{\boldsymbol{y} \in \mathcal{P}'} g(\boldsymbol{y})$, where $\mathcal{P}' = \{\boldsymbol{y} \in \mathbb{R}^n | h_i(\boldsymbol{y}) \leq b_i, \forall i \in [m], \boldsymbol{y} \succeq_{\mathcal{K}_{\boldsymbol{\alpha}}} \boldsymbol{0}\}$ is down-closed w.r.t. the conic inequality $\preceq_{\mathcal{K}_{\boldsymbol{\alpha}}}$. The down-closedness here means if $\boldsymbol{a} \in \mathcal{P}'$ and $\boldsymbol{0} \preceq_{\mathcal{K}_{\boldsymbol{\alpha}}} \boldsymbol{b} \preceq_{\mathcal{K}_{\boldsymbol{\alpha}}} \boldsymbol{a}$, then $\boldsymbol{b} \in \mathcal{P}'$ as well.

Let $\mathbf{A} := \mathrm{diag}(\boldsymbol{\alpha})$, and a function $f(\boldsymbol{x}) := g(\mathbf{A}\boldsymbol{x})$. One can see that if $g$ is $\mathcal{K}_{\boldsymbol{\alpha}}$-DR-submodular, then $f$ is DR-submodular: assume wlog.[3] that $g$ is twice differentiable, then $\nabla^2 f(\boldsymbol{x}) = \mathbf{A}^\top \nabla^2 g \mathbf{A}$, and $\nabla_{ij}^2 f(\boldsymbol{x}) = \alpha_i \alpha_j \nabla_{ij}^2 g \leq 0$, so $f$ is DR-submodular.

By the affine transformation $\boldsymbol{y} := \mathbf{A}\boldsymbol{x}$, one can transform the $\mathcal{K}_{\boldsymbol{\alpha}}$-DR-submodular maximization problem to be a DR-submodular maximization problem $\max_{\boldsymbol{x} \in \mathcal{P}} g(\mathbf{A}\boldsymbol{x})$, where $\mathcal{P} = \{\boldsymbol{x} \in \mathbb{R}^n | h_i(\mathbf{A}\boldsymbol{x}) \leq b_i, \forall i \in [m], \mathbf{A}\boldsymbol{x} \succeq_{\mathcal{K}_{\boldsymbol{\alpha}}} \boldsymbol{0}\}$ is down-closed w.r.t. the ordinary component-wise inequality $\leq$. To verify the down-closedness of $\mathcal{P}$ w.r.t. to the ordinary inequality $\leq$ here, let $\boldsymbol{y}_1 = \mathbf{A}\boldsymbol{x}_1 \in \mathcal{P}'$ (so $\boldsymbol{x}_1 \in \mathcal{P}$). Suppose there is a point $\boldsymbol{y}_2 = \mathbf{A}\boldsymbol{x}_2$ s.t. $\boldsymbol{0} \preceq_{\mathcal{K}_{\boldsymbol{\alpha}}} \boldsymbol{y}_2 \preceq_{\mathcal{K}_{\boldsymbol{\alpha}}} \boldsymbol{y}_1$. From the down-closedness of $\mathcal{P}'$, we know that $\boldsymbol{y}_2 \in \mathcal{P}'$, thus $\boldsymbol{x}_2 \in \mathcal{P}$. Looking at $\boldsymbol{0} \preceq_{\mathcal{K}_{\boldsymbol{\alpha}}} \boldsymbol{y}_2 \preceq_{\mathcal{K}_{\boldsymbol{\alpha}}} \boldsymbol{y}_1$, it is equivalent to that $0 \leq \boldsymbol{x}_2 \leq \boldsymbol{x}_1$. Thus we establish the down-closedness of $\mathcal{P}$.

Given the reduction, we can reuse the algorithms for the original DR-submodular maximization problem (P).

## A.3 Proof for the Logistic Loss in Section 2.1

Remember that the logistic loss is:

$$l(\boldsymbol{x}) = \frac{1}{m} \sum_{j=1}^m f_j(\boldsymbol{x}) = \frac{1}{m} \sum_{j=1}^m \log(1 + \exp(-y_j \boldsymbol{x}^\top \boldsymbol{z}^j)) \tag{17}$$

**Claim 3.** *$l(\boldsymbol{x})$ in (17) is $\mathcal{K}_{\boldsymbol{\alpha}}$-DR-supermodular.*

*Proof of Claim 3.* To show that $l(\boldsymbol{x})$ is $\mathcal{K}_{\boldsymbol{\alpha}}$-DR-supermodular, we can check the second-order condition in (16), that is, whether it holds that $\alpha_p \alpha_q \nabla_{pq}^2 l(\boldsymbol{x}) \geq 0, \ \forall p, q \in [n]$. One can easily see

that,

$$\frac{\partial l(\boldsymbol{x})}{\partial x_p} = \frac{1}{m} \sum\nolimits_{j=1}^m \frac{-y_j z_p^j}{\exp\left(y_j \boldsymbol{x}^\top \boldsymbol{z}^j\right) + 1}$$

$$\frac{\partial^2 l(\boldsymbol{x})}{\partial x_p \partial x_q} = \frac{1}{m} \sum\nolimits_{j=1}^m \frac{\exp\left(y_j \boldsymbol{x}^\top \boldsymbol{z}^j\right)}{[\exp\left(y_j \boldsymbol{x}^\top \boldsymbol{z}^j\right) + 1]^2} z_p^j z_q^j.$$

Since $\alpha_p = \text{sign}(z_p^j)$, so $\alpha_p \alpha_q \nabla_{pq}^2 l(\boldsymbol{x}) \geq 0$, $\forall p, q \in [n]$. Thus $l(\boldsymbol{x})$ in (17) is $\mathcal{K}_{\boldsymbol{\alpha}}$-DR-supermodular according to (16). ∎

## B More Applications

We present more applications that fall into submodular or $\mathcal{K}_{\boldsymbol{\alpha}}$-submodular optimization problems. One class of notable examples are the objectives studied in [10] in the online setting. These objectives are captured by the DR-submodular property over continuous domains. One can also refer to Section 2.2 in [3] to see more examples.

**DR-submodular quadratic functions.** Price optimization with continuous prices is a DR-submodular quadratic optimization problem [22]. Another representative class of DR-submodular quadratic objectives arises when computing the stability number $s(G)$ of a graph $G = (V, E)$ [30], $s(G)^{-1} = \min_{\boldsymbol{x} \in \Delta} \boldsymbol{x}^\top (\mathbf{A} + \mathbf{I}) \boldsymbol{x}$, where $\mathbf{A}$ is the adjacency matrix of the graph $G$, $\Delta$ is the standard simplex. This is an instance of a convex-constrained DR-submodular maximization problem.

**Non-negative PCA (NN-PCA).** NN-PCA [40; 29] is widely used as alternative models of PCA for dimension reduction, since its projection involves only non-negative weights—a required property in fields like economics, bioinformatics and computer vision. For a given set of $m$ data points $\boldsymbol{z}^j \in \mathbb{R}^n, j \in [m]$, NN-PCA aims to solve the following non-convex optimization problem:

$$\min_{\|\boldsymbol{x}\|_2 \leq 1, \boldsymbol{x} \geq 0} f(\boldsymbol{x}) := -\frac{1}{2} \boldsymbol{x}^\top \left(\sum\nolimits_{j=1}^m \boldsymbol{z}^j {\boldsymbol{z}^j}^\top\right) \boldsymbol{x}. \tag{18}$$

Let $\mathbf{A} = \sum_{j=1}^m \boldsymbol{z}^j {\boldsymbol{z}^j}^\top$, one can see that,

$$A_{pp} = \sum\nolimits_{j=1}^m (z_p^j)^2 \geq 0, \ A_{pq} = \sum\nolimits_{j=1}^m z_p^j z_q^j = A_{qp}.$$

Let us make the following weak assumption: For one dimension/feature $i$, all the data points have the same sign, i.e., $\text{sign}(z_i^j)$ is the same for all $j \in [m]$ (which can be achieved by easily scaling if not). Now, by choosing the sign vector $\boldsymbol{\alpha} \in \{\pm 1\}^n$ to be $\alpha_p = \text{sign}(z_p^j), \forall p \in [n]$, one can easily verify that $A_{pq} \alpha_p \alpha_q \geq 0, \forall p, q \in [n]$. Notice that $\nabla^2 f$ in (18) is $-\mathbf{A}$, so it holds that $\alpha_p \alpha_q \nabla_{pq}^2 f \leq 0, \forall p, q \in [n]$, thus $f(\boldsymbol{x})$ is $\mathcal{K}_{\boldsymbol{\alpha}}$-DR-submodular according to (16). Thus we can treat (18) as a constrained $\mathcal{K}_{\boldsymbol{\alpha}}$-DR-submodular minimization problem.

**Submodular spectral functions.** As discussed by [3], submodular spectral functions [14] in the following form are DR-submodular,

$$f(\boldsymbol{x}) = \log \det \left(\sum\nolimits_{i=1}^n x_i \mathbf{A}_i\right), \boldsymbol{x} \in \mathbb{R}_+^n, \tag{19}$$

where $\mathbf{A}_i$ are positive definite matrices. One can check the DR-submodularity of $f(\boldsymbol{x})$ by checking its second-order-derivatives.

## C The Subroutine Algorithm

Algorithm 3 is taken from [27], the only difference lies in the output: we output the solution $\boldsymbol{x}^{(k')}$ with the minimum non-stationarity, which is needed to apply the local-global relation. While [27] outputs the solution in the last step. Since $C_f(\mathcal{P})$ is generally hard to evaluate, we used the classical FRANK-WOLFE step size $\frac{2}{k+2}$ in the experiments.

**Algorithm 3:** NON-CONVEX FRANK-WOLFE $(f, \mathcal{P}, K, \epsilon, \boldsymbol{x}^{(0)})$ [27]

> **Input**: $\max_{\boldsymbol{x} \in \mathcal{P}} f(\boldsymbol{x})$, $\mathcal{P}$: convex set, $K$: number of iterations, $\epsilon$: stopping tolerance
> **1 for** $k = 0, ..., K$ **do**
> **2**    find $\boldsymbol{v}^{(k)}$ s.t. $\langle \boldsymbol{v}^{(k)}, \nabla f(\boldsymbol{x}^{(k)}) \rangle \geq \max_{\boldsymbol{v} \in \mathcal{P}} \langle \boldsymbol{v}, \nabla f(\boldsymbol{x}^{(k)}) \rangle$;      // LMO
> **3**    $\boldsymbol{d}_k \leftarrow \boldsymbol{v}^{(k)} - \boldsymbol{x}^{(k)}$, $g_k := \langle \boldsymbol{d}_k, \nabla f(\boldsymbol{x}^{(k)}) \rangle$ ;      // $g_k$: non-stationarity measure
> **4**    **if** $g_k \leq \epsilon$ **then return** $\boldsymbol{x}^{(k)}$;
> **5**    Option I: $\gamma_k \in \arg\min_{\gamma \in [0,1]} f(\boldsymbol{x}^{(k)} + \gamma \boldsymbol{d}^{(k)})$, Option II: $\gamma_k \leftarrow \min\{\frac{g_k}{C}, 1\}$ for $C \geq C_f(\mathcal{P})$ ;
> **6**    $\boldsymbol{x}^{(k+1)} \leftarrow \boldsymbol{x}^{(k)} + \gamma_k \boldsymbol{d}^{(k)}$ ;
>
> **Output**: $\boldsymbol{x}^{(k')}$ and $g_{k'} = \min_{0 \leq k \leq K} g_k$ ;      // modified output solution compared to [27]

## D Proofs for Properties

### D.1 Proof of Lemma 1

*Proof of Lemma 1.* Since $f$ is DR-submodular, so it is concave along any direction $\boldsymbol{v} \in \pm\mathbb{R}_+^n$. We know that $\boldsymbol{x} \vee \boldsymbol{y} - \boldsymbol{x} \geq 0$ and $\boldsymbol{x} \wedge \boldsymbol{y} - \boldsymbol{x} \leq 0$, so from the strong DR-submodularity in (6),

$$f(\boldsymbol{x} \vee \boldsymbol{y}) - f(\boldsymbol{x}) \leq \langle \nabla f(\boldsymbol{x}), \boldsymbol{x} \vee \boldsymbol{y} - \boldsymbol{x} \rangle - \frac{\mu}{2} \|\boldsymbol{x} \vee \boldsymbol{y} - \boldsymbol{x}\|^2,$$

$$f(\boldsymbol{x} \wedge \boldsymbol{y}) - f(\boldsymbol{x}) \leq \langle \nabla f(\boldsymbol{x}), \boldsymbol{x} \wedge \boldsymbol{y} - \boldsymbol{x} \rangle - \frac{\mu}{2} \|\boldsymbol{x} \wedge \boldsymbol{y} - \boldsymbol{x}\|^2.$$

Summing the above two inequalities and notice that $\boldsymbol{x} \vee \boldsymbol{y} + \boldsymbol{x} \wedge \boldsymbol{y} = \boldsymbol{x} + \boldsymbol{y}$, we arrive,

$$(\boldsymbol{y} - \boldsymbol{x})^\top \nabla f(\boldsymbol{x}) \geq f(\boldsymbol{x} \vee \boldsymbol{y}) + f(\boldsymbol{x} \wedge \boldsymbol{y}) - 2f(\boldsymbol{x}) + \frac{\mu}{2}(\|\boldsymbol{x} \vee \boldsymbol{y} - \boldsymbol{x}\|^2 + \|\boldsymbol{x} \wedge \boldsymbol{y} - \boldsymbol{x}\|^2)$$

$$= f(\boldsymbol{x} \vee \boldsymbol{y}) + f(\boldsymbol{x} \wedge \boldsymbol{y}) - 2f(\boldsymbol{x}) + \frac{\mu}{2} \|\boldsymbol{y} - \boldsymbol{x}\|^2,$$

the last equality holds since $\|\boldsymbol{x} \vee \boldsymbol{y} - \boldsymbol{x}\|^2 + \|\boldsymbol{x} \wedge \boldsymbol{y} - \boldsymbol{x}\|^2 = \|\boldsymbol{y} - \boldsymbol{x}\|^2$. ∎

### D.2 Proof of Proposition 1

*Proof of Proposition 1.* Consider the point $\boldsymbol{z}^* := \boldsymbol{x} \vee \boldsymbol{x}^* - \boldsymbol{x} = (\boldsymbol{x}^* - \boldsymbol{x}) \vee 0$. One can see that: 1) $0 \leq \boldsymbol{z}^* \leq \boldsymbol{x}^*$; 2) $\boldsymbol{z}^* \in \mathcal{P}$ (down-closedness); 3) $\boldsymbol{z}^* \in \mathcal{Q}$ (because of $\boldsymbol{z}^* \leq \bar{\boldsymbol{u}} - \boldsymbol{x}$). From Lemma 1,

$$\langle \boldsymbol{x}^* - \boldsymbol{x}, \nabla f(\boldsymbol{x}) \rangle + 2f(\boldsymbol{x}) \geq f(\boldsymbol{x} \vee \boldsymbol{x}^*) + f(\boldsymbol{x} \wedge \boldsymbol{x}^*) + \frac{\mu}{2} \|\boldsymbol{x} - \boldsymbol{x}^*\|^2, \tag{20}$$

$$\langle \boldsymbol{z}^* - \boldsymbol{z}, \nabla f(\boldsymbol{z}) \rangle + 2f(\boldsymbol{z}) \geq f(\boldsymbol{z} \vee \boldsymbol{z}^*) + f(\boldsymbol{z} \wedge \boldsymbol{z}^*) + \frac{\mu}{2} \|\boldsymbol{z} - \boldsymbol{z}^*\|^2. \tag{21}$$

Let us first of all prove the following key Claim.

**Claim 1.** *It holds that* $f(\boldsymbol{x} \vee \boldsymbol{x}^*) + f(\boldsymbol{x} \wedge \boldsymbol{x}^*) + f(\boldsymbol{z} \vee \boldsymbol{z}^*) + f(\boldsymbol{z} \wedge \boldsymbol{z}^*) \geq f(\boldsymbol{x}^*)$.

*Proof of Claim 1.* Firstly, we are going to prove that

$$f(\boldsymbol{x} \vee \boldsymbol{x}^*) + f(\boldsymbol{z} \vee \boldsymbol{z}^*) \geq f(\boldsymbol{z}^*) + f((\boldsymbol{x} + \boldsymbol{z}) \vee \boldsymbol{x}^*), \tag{22}$$

which is equivalent to $f(\boldsymbol{x} \vee \boldsymbol{x}^*) - f(\boldsymbol{z}^*) \geq f((\boldsymbol{x} + \boldsymbol{z}) \vee \boldsymbol{x}^*) - f(\boldsymbol{z} \vee \boldsymbol{z}^*)$. It can be shown that $\boldsymbol{x} \vee \boldsymbol{x}^* - \boldsymbol{z}^* = (\boldsymbol{x} + \boldsymbol{z}) \vee \boldsymbol{x}^* - \boldsymbol{z} \vee \boldsymbol{z}^*$. Combining this with the fact that $\boldsymbol{z}^* \leq \boldsymbol{z} \vee \boldsymbol{z}^*$, and using the DR property (see (1)) implies (22). Then we establish,

$$\boldsymbol{x} \vee \boldsymbol{x}^* - \boldsymbol{z}^* = (\boldsymbol{x} + \boldsymbol{z}) \vee \boldsymbol{x}^* - \boldsymbol{z} \vee \boldsymbol{z}^* . \tag{23}$$

We will show that both the RHS and LHS of the above equation are equal to $\boldsymbol{x}$: for the LHS of (23) we can write $\boldsymbol{x} \vee \boldsymbol{x}^* - \boldsymbol{z}^* = \boldsymbol{x} \vee \boldsymbol{x}^* - (\boldsymbol{x} \vee \boldsymbol{x}^* - \boldsymbol{x}) = \boldsymbol{x}$. For the RHS of (23) let us consider any coordinate $i \in [n]$,

$$(x_i + z_i) \vee x_i^* - z_i \vee z_i^* = (x_i + z_i) \vee x_i^* - ((x_i + z_i) - x_i) \vee ((x_i \vee x_i^*) - x_i) = x_i,$$

where the last equality holds easily for the two situations: $(x_i + z_i) \geq x_i^*$ and $(x_i + z_i) < x_i^*$.

Next, we are going to prove that,

$$f(\boldsymbol{z}^*) + f(\boldsymbol{x} \wedge \boldsymbol{x}^*) \geq f(\boldsymbol{x}^*) + f(0) \tag{24}$$

it is equivalent to $f(\boldsymbol{z}^*) - f(0) \geq f(\boldsymbol{x}^*) - f(\boldsymbol{x} \wedge \boldsymbol{x}^*)$, which can be done similarly by the DR property: Notice that

$$\boldsymbol{x}^* - \boldsymbol{x} \wedge \boldsymbol{x}^* = \boldsymbol{x} \vee \boldsymbol{x}^* - \boldsymbol{x} = \boldsymbol{z}^* - 0 \text{ and } 0 \leq \boldsymbol{x} \wedge \boldsymbol{x}^*$$

thus (24) holds from the DR property. Combining (22) and (24) one can get,

$$f(\boldsymbol{x} \vee \boldsymbol{x}^*) + f(\boldsymbol{z} \vee \boldsymbol{z}^*) + f(\boldsymbol{x} \wedge \boldsymbol{x}^*) + f(\boldsymbol{z} \wedge \boldsymbol{z}^*) \geq f(\boldsymbol{x}^*) + f(0) + f((\boldsymbol{x} + \boldsymbol{z}) \vee \boldsymbol{x}^*) + f(\boldsymbol{z} \wedge \boldsymbol{z}^*)$$
$$\geq f(\boldsymbol{x}^*) \quad \text{(non-negativity of } f) \,.$$

$\blacksquare$

Combining (20) and (21) and Claim 1 it reads,

$$\langle \boldsymbol{x}^* - \boldsymbol{x}, \nabla f(\boldsymbol{x}) \rangle + \langle \boldsymbol{z}^* - \boldsymbol{z}, \nabla f(\boldsymbol{z}) \rangle + 2(f(\boldsymbol{x}) + f(\boldsymbol{z})) \geq f(\boldsymbol{x}^*) + \frac{\mu}{2}(\|\boldsymbol{x} - \boldsymbol{x}^*\|^2 + \|\boldsymbol{z} - \boldsymbol{z}^*\|^2) \tag{25}$$

From the definition of non-stationarity in (8) one can get,

$$g_{\mathcal{P}}(\boldsymbol{x}) := \max_{\boldsymbol{v} \in \mathcal{P}} \langle \boldsymbol{v} - \boldsymbol{x}, \nabla f(\boldsymbol{x}) \rangle \overset{\boldsymbol{x}^* \in \mathcal{P}}{\geq} \langle \boldsymbol{x}^* - \boldsymbol{x}, \nabla f(\boldsymbol{x}) \rangle \tag{26}$$

$$g_{\mathcal{Q}}(\boldsymbol{z}) := \max_{\boldsymbol{v} \in \mathcal{Q}} \langle \boldsymbol{v} - \boldsymbol{z}, \nabla f(\boldsymbol{z}) \rangle \overset{\boldsymbol{z}^* \in \mathcal{Q}}{\geq} \langle \boldsymbol{z}^* - \boldsymbol{z}, \nabla f(\boldsymbol{z}) \rangle \tag{27}$$

Putting together (25) to (27) we can get,

$$2(f(\boldsymbol{x}) + f(\boldsymbol{z})) \geq f(\boldsymbol{x}^*) - g_{\mathcal{P}}(\boldsymbol{x}) - g_{\mathcal{Q}}(\boldsymbol{z}) + \frac{\mu}{2}(\|\boldsymbol{x} - \boldsymbol{x}^*\|^2 + \|\boldsymbol{z} - \boldsymbol{z}^*\|^2).$$

so it arrives $\max\{f(\boldsymbol{x}), f(\boldsymbol{z})\} \geq \frac{1}{4}[f(\boldsymbol{x}^*) - g_{\mathcal{P}}(\boldsymbol{x}) - g_{\mathcal{Q}}(\boldsymbol{z})] + \frac{\mu}{8}(\|\boldsymbol{x} - \boldsymbol{x}^*\|^2 + \|\boldsymbol{z} - \boldsymbol{z}^*\|^2).$ $\blacksquare$

# E  Proofs for Algorithms

## E.1  Proof of Theorem 1

*Proof of Theorem 1.* Let $g_{\mathcal{P}}(\boldsymbol{x}), g_{\mathcal{Q}}(\boldsymbol{z})$ to be the non-stationarity of $\boldsymbol{x}$ and $\boldsymbol{z}$, respectively. Since we are using the NON-CONVEX FRANK-WOLFE (Algorithm 3) as subroutine, according to [27, Theorem 1], one can get,

$$g_{\mathcal{P}}(\boldsymbol{x}) \leq \min \left\{ \frac{\max\{2h_1, C_f(\mathcal{P})\}}{\sqrt{K_1 + 1}}, \epsilon_1 \right\}$$

$$g_{\mathcal{Q}}(\boldsymbol{z}) \leq \min \left\{ \frac{\max\{2h_2, C_f(\mathcal{Q})\}}{\sqrt{K_2 + 1}}, \epsilon_2 \right\},$$

Plugging the above into Proposition 1 we reach the conclusion in (10). $\blacksquare$

## E.2  Proof of Lemma 2

**Lemma 2.** *Assume $\boldsymbol{x}^{(0)} = 0$. For $k = 0, ..., K - 1$, it holds $x_i^{(k)} \leq \bar{u}_i[1 - (1 - \gamma)^{t^{(k)}/\gamma}], \forall i \in [n]$.*

*Proof of Lemma 2.* We prove by induction. First of all, it holds when $k = 0$, since $x_i^{(0)} = 0$, and $t^{(0)} = 0$ as well. Assume it holds for $k$. Then for $k + 1$, we have

$$
\begin{aligned}
x_i^{(k+1)} &= x_i^{(k)} + \gamma v_i^{(k)} \\
&\leq x_i^{(k)} + \gamma(\bar{u}_i - x_i^{(k)}) \quad \text{(constraint of shrunken LMO)} \tag{28} \\
&= (1 - \gamma)x_i^{(k)} + \gamma \bar{u}_i \\
&\leq (1 - \gamma)\bar{u}_i[1 - (1 - \gamma)^{t^{(k)}/\gamma}] + \gamma \bar{u}_i \quad \text{(induction)} \tag{29} \\
&= \bar{u}_i[1 - (1 - \gamma)^{t^{(k+1)}/\gamma}].
\end{aligned}
$$

$\blacksquare$

### E.3 Proof of Lemma 3

**Lemma 3** (Generalized from Lemma 7 in [8]). *Given $\boldsymbol{\theta} \in (0, \bar{\boldsymbol{u}}]$, let $\lambda' = \min_{i \in [n]} \frac{\bar{u}_i}{\theta_i}$. Then for all $\boldsymbol{x} \in [0, \boldsymbol{\theta}]$, it holds $f(\boldsymbol{x} \vee \boldsymbol{x}^*) \geq (1 - \frac{1}{\lambda'})f(\boldsymbol{x}^*)$.*

*Proof of Lemma 3.* Consider $r(\lambda) = \boldsymbol{x}^* + \lambda(\boldsymbol{x} \vee \boldsymbol{x}^* - \boldsymbol{x}^*)$, it is easy to see that $r(\lambda) \geq 0, \forall \lambda \geq 0$.

Notice that $\lambda' \geq 1$. Let $\boldsymbol{y} = r(\lambda') = \boldsymbol{x}^* + \lambda'(\boldsymbol{x} \vee \boldsymbol{x}^* - \boldsymbol{x}^*)$, it is easy to see that $\boldsymbol{y} \geq 0$, it also hold that $\boldsymbol{y} \leq \bar{\boldsymbol{u}}$: Consider one coordinate $i$, 1) if $x_i \geq x_i^*$, then $y_i = x_i^* + \lambda'(x_i - x_i^*) \leq \lambda' x_i \leq \lambda' \theta_i \leq \bar{u}_i$; 2) if $x_i < x_i^*$, then $y_i = x_i^* \leq \bar{u}_i$. So $f(\boldsymbol{y}) \geq 0$.

Note that

$$\boldsymbol{x} \vee \boldsymbol{x}^* = (1 - \frac{1}{\lambda'})\boldsymbol{x}^* + \frac{1}{\lambda'}\boldsymbol{y} = (1 - \frac{1}{\lambda'})r(0) + \frac{1}{\lambda'}r(\lambda'),$$

since $f$ is concave along $r(\lambda)$, so it holds that,

$$f(\boldsymbol{x} \vee \boldsymbol{x}^*) \geq (1 - \frac{1}{\lambda'})f(\boldsymbol{x}^*) + \frac{1}{\lambda'}f(\boldsymbol{y}) \geq (1 - \frac{1}{\lambda'})f(\boldsymbol{x}^*).$$

∎

### E.4 Proof of Theorem 2

*Proof of Theorem 2.* First of all, let us prove the Claim:

**Claim 2.** *For $k = 0, ..., K-1$ it holds $f(\boldsymbol{x}^{(k+1)}) \geq (1-\gamma)f(\boldsymbol{x}^{(k)}) + \gamma(1-\gamma)^{t^{(k)}/\gamma}f(\boldsymbol{x}^*) - \frac{LD^2}{2}\gamma^2$,*

*Proof of Claim 2.* Consider a point $\boldsymbol{z}^{(k)} := \boldsymbol{x}^{(k)} \vee \boldsymbol{x}^* - \boldsymbol{x}^{(k)}$, one can observe that: 1) $\boldsymbol{z}^{(k)} \leq \bar{\boldsymbol{u}} - \boldsymbol{x}^{(k)}$; 2) since $\boldsymbol{x}^{(k)} \geq 0, \boldsymbol{x}^* \geq 0$, so $\boldsymbol{z}^{(k)} \leq \boldsymbol{x}^*$, which implies that $\boldsymbol{z}^{(k)} \in \mathcal{P}$ (from down-closedness of $\mathcal{P}$). So $\boldsymbol{z}^{(k)}$ is a candidate solution for the new LMO (Step 3). We have,

$$
\begin{aligned}
f(\boldsymbol{x}^{(k+1)}) - f(\boldsymbol{x}^{(k)}) &\geq \gamma\langle\nabla f(\boldsymbol{x}^{(k)}), \boldsymbol{v}^{(k)}\rangle - \frac{L}{2}\gamma^2\|\boldsymbol{v}^{(k)}\|^2 \quad \text{(Quadratic lower bound from (5))}\\
&\geq \gamma\langle\nabla f(\boldsymbol{x}^{(k)}), \boldsymbol{v}^{(k)}\rangle - \frac{L}{2}\gamma^2 D^2 \quad \text{(diameter of } \mathcal{P})\\
&\geq \gamma\langle\nabla f(\boldsymbol{x}^{(k)}), \boldsymbol{z}^{(k)}\rangle - \frac{L}{2}\gamma^2 D^2 \quad \text{(shrunken LMO)}\\
&\geq \gamma(f(\boldsymbol{x}^{(k)} + \boldsymbol{z}^{(k)}) - f(\boldsymbol{x}^{(k)})) - \frac{L}{2}\gamma^2 D^2 \quad \text{(concave along } \boldsymbol{z}^{(k)})\\
&= \gamma[f(\boldsymbol{x}^{(k)} \vee \boldsymbol{x}^*) - f(\boldsymbol{x}^{(k)})] - \frac{L}{2}\gamma^2 D^2\\
&\geq \gamma[(1 - \frac{1}{\lambda'})f(\boldsymbol{x}^*) - f(\boldsymbol{x}^{(k)})] - \frac{L}{2}\gamma^2 D^2 \quad \text{(Lemma 3)}\\
&= \gamma[(1-\gamma)^{t^{(k)}/\gamma}f(\boldsymbol{x}^*) - f(\boldsymbol{x}^{(k)})] - \frac{L}{2}\gamma^2 D^2
\end{aligned}
$$

where the last equality comes from setting $\boldsymbol{\theta} := \bar{\boldsymbol{u}}(1 - (1-\gamma)^{t^{(k)}/\gamma})$ according to Lemma 2, thus $\lambda' = \min_i \frac{\bar{u}_i}{\theta_i} = (1 - (1-\gamma)^{t^{(k)}/\gamma})^{-1}$.

After rearrangement, we reach the claim. ∎

Then, let us prove Theorem 2 by *induction*.

First of all, it holds when $k = 0$ (notice that $t^{(0)} = 0$). Assume that it holds for $k$. Then for $k + 1$, considering the fact $e^{-t} - O(\gamma) \leq (1-\gamma)^{t/\gamma}$ when $0 < \gamma \leq t \leq 1$ and Claim 2 we get,

$$f(\boldsymbol{x}^{(k+1)})$$

$$\geq (1-\gamma)f(\boldsymbol{x}^{(k)}) + \gamma(1-\gamma)^{t^{(k)}/\gamma}f(\boldsymbol{x}^*) - \frac{LD^2}{2}\gamma^2$$

$$\geq (1-\gamma)f(\boldsymbol{x}^{(k)}) + \gamma[e^{-t^{(k)}} - O(\gamma)]f(\boldsymbol{x}^*) - \frac{LD^2}{2}\gamma^2 \tag{30}$$

$$\geq (1-\gamma)[t^{(k)}e^{-t^{(k)}}f(\boldsymbol{x}^*) - \frac{LD^2}{2}k\gamma^2 - O(\gamma^2)f(\boldsymbol{x}^*)] + \gamma[e^{-t^{(k)}} - O(\gamma)]f(\boldsymbol{x}^*) - \frac{LD^2}{2}\gamma^2$$

$$= [(1-\gamma)t^{(k)}e^{-t^{(k)}} + \gamma e^{-t^{(k)}}]f(\boldsymbol{x}^*) - \frac{LD^2}{2}\gamma^2[(1-\gamma)k+1] - [(1-\gamma)O(\gamma^2) + \gamma O(\gamma)]f(\boldsymbol{x}^*)$$

$$\geq [(1-\gamma)t^{(k)}e^{-t^{(k)}} + \gamma e^{-t^{(k)}}]f(\boldsymbol{x}^*) - \frac{LD^2}{2}\gamma^2(k+1) - O(\gamma^2)f(\boldsymbol{x}^*). \tag{31}$$

Let us consider the term $[(1-\gamma)t^{(k)}e^{-t^{(k)}} + \gamma e^{-t^{(k)}}]f(\boldsymbol{x}^*)$. We know that the function $g(t) = te^{-t}$ is concave in $[0,2]$, so $g(t^{(k)} + \gamma) - g(t^{(k)}) \leq \gamma g'(t^{(k)})$, which amounts to

$$[(1-\gamma)t^{(k)}e^{-t^{(k)}} + \gamma e^{-t^{(k)}}]f(\boldsymbol{x}^*) \geq (t^{(k)} + \gamma)e^{-(t^{(k)}+\gamma)}f(\boldsymbol{x}^*) = t^{(k+1)}e^{-t^{(k+1)}}f(\boldsymbol{x}^*) \tag{32}$$

Plugging (32) into (31) we get,

$$f(\boldsymbol{x}^{(k+1)}) \geq t^{(k+1)}e^{-t^{(k+1)}}f(\boldsymbol{x}^*) - \frac{LD^2}{2}\gamma^2(k+1) - O(\gamma^2)f(\boldsymbol{x}^*).$$

Thus proving the induction, and proving the theorem as well. ∎

## F  Miscellaneous Results

### F.1  Verifying DR-submodularity of the Objectives

**Softmax extension.** For softmax extension, the objective is,

$$f(\boldsymbol{x}) = \log\det(\operatorname{diag}(\boldsymbol{x})(\mathbf{L} - \mathbf{I}) + \mathbf{I}), \boldsymbol{x} \in [0,1]^n.$$

Its DR-submodularity can be established by directly applying Lemma 3 in [20]: [20, Lemma 3] immediately implies that all entries of $\nabla^2 f$ are non-positive, so $f(\boldsymbol{x})$ is DR-submodular.

**Multilinear extension.** The DR-submodularity of multilinear extension can be directly recognized by considering the conclusion in Appendix A.2 of [3] and the fact that multilinear extension is coordinate-wise linear.

**KL$(\boldsymbol{x})$.** The Kullback-Leibler divergence between $q_{\boldsymbol{x}}$ and $p$, i.e., $\sum_{S \subseteq \mathcal{V}} q_{\boldsymbol{x}}(S) \log \frac{q_{\boldsymbol{x}}(S)}{p(S)}$ is,

$$\text{KL}(\boldsymbol{x}) = -\sum_{S \subseteq \mathcal{V}} \prod_{i \in S} x_i \prod_{j \notin S}(1-x_j)F(S) + \sum_{i=1}^{n}[x_i \log x_i + (1-x_i)\log(1-x_i)] + \log Z.$$

The first term is the negative of a multilinear extension, so it is DR-supermodular. The second term is separable, and coordinate-wise convex, so it will not affect the off-diagonal entries of $\nabla^2\text{KL}(\boldsymbol{x})$, it will only contribute to the diagonal entries. Now, one can see that all entries of $\nabla^2\text{KL}(\boldsymbol{x})$ are non-negative, so KL$(\boldsymbol{x})$ is DR-supermodular w.r.t. $\boldsymbol{x}$.

### F.2  A Counter Example to Show PSD Cone is not Lattice

The positive semidefine cone $\mathcal{K}_{\text{PSD}} = \{\mathbf{A} \in \mathbb{R}^{n \times n} | \mathbf{A} \text{ is symmetric}, \mathbf{A} \succeq 0\}$ is a proper cone, but not a lattice cone. That is, it can not be used to define a lattice over the space of symmetric matrices.

Let us consider the two dimensional symmetric matrix space $S^2$. Specifically, the following two symmetric matrices,

$$\mathbf{X} = \begin{bmatrix} 1 & 0 \\ 0 & 0 \end{bmatrix}, \mathbf{Y} = \begin{bmatrix} 0 & 0 \\ 0 & 1 \end{bmatrix}.$$

For the conic inequality $\preceq_{\mathcal{K}_{\text{PSD}}}$, assume that there exists a least upper bound, i.e., the join of $\mathbf{X}, \mathbf{Y}$: $\mathbf{Z} := \mathbf{X} \vee \mathbf{Y}$. From the definition of least upper bound, $\forall \mathbf{W} \in S^2$ it should hold that,

$$\mathbf{W} \succeq_{\mathcal{K}_{\text{PSD}}} \mathbf{X} \text{ and } \mathbf{W} \succeq_{\mathcal{K}_{\text{PSD}}} \mathbf{Y} \text{ iff } \mathbf{W} \succeq_{\mathcal{K}_{\text{PSD}}} \mathbf{Z}. \tag{33}$$

Suppose $\mathbf{Z} = \begin{bmatrix} b & a \\ a & c \end{bmatrix}$. Firstly, consider $\mathbf{W}$ to be diagonal matrices, one can verify that $\mathbf{Z}$ must be in the form of $\begin{bmatrix} 1 & a \\ a & 1 \end{bmatrix}$, then considering $\mathbf{W} = \mathbf{I}$ forcing $\mathbf{Z}$ to be $\mathbf{I}$.

Now let $\mathbf{W} = \frac{2}{3} \begin{bmatrix} 2 & 1 \\ 1 & 2 \end{bmatrix}$, which is $\succeq_{\mathcal{K}_{\text{PSD}}} \mathbf{X}$ and $\succeq_{\mathcal{K}_{\text{PSD}}} \mathbf{Y}$. However, $\mathbf{W} - \mathbf{I} = \frac{1}{3} \begin{bmatrix} 1 & 2 \\ 2 & 1 \end{bmatrix} \notin \mathcal{K}_{\text{PSD}}$, thus contradicting (33).

# G   Additional Experimental Results

We generate the down-closed polytope constraints in the same form and same way as that for DR-submodular quadratic functions.

Figure 5 shows the function values returned by different solvers w.r.t. $n$, for which the random polytope constraints were generated with exponential distribution. Specifically, the random polytope is in the form of $\mathcal{P} = \{\boldsymbol{x} \in \mathbb{R}_+^n \mid \mathbf{A}\boldsymbol{x} \leq \boldsymbol{b}, \boldsymbol{x} \leq \bar{\boldsymbol{u}}, \mathbf{A} \in \mathbb{R}_{++}^{m \times n}, \boldsymbol{b} \in \mathbb{R}_+^m\}$. Each entry of $\mathbf{A}$ was sampled from $\text{Exp}(1) + \nu$, where $\nu = 0.01$ is a small positive constant. We set $\boldsymbol{b} = 2 * \mathbf{1}^m$, and set $\bar{\boldsymbol{u}}$ to be the tightest upper bound of $\mathcal{P}$ by $\bar{u}_j = \min_{i \in [m]} \frac{b_i}{A_{ij}}, \forall j \in [n]$. One can see that the TWO-PHASE FRANK-WOLFE has the best performance, while non-monotone FRANK-WOLFE and PROJGRAD have comparable performance.

Figure 5: Results on softmax instances with random polytope constraints generated from exponential distribution.

## Footnotes

[3]If not one can still use other equivalent characterizations, for instance, the characterization in (12) or in (13) to formulate this.