[Reviews · NeurIPS 2017]

Reviewer 1



For set functions, submodularity is equivalent to diminishing returns (DR), however, for integer-lattice functions or continuous functions, DR-submodular functions form an important subclass of submodular functions, and their maximization captures non-convex problems of interest such as maximizing the softmax extension of DPPs. In this paper, the authors study non-monotone continuous DR-submodular maximization under general down-closed convex constraints. They prove a relation between approximate stationary points and the global optimum which enables them to use recent results on non-convex Frank-Wolfe to devise a two-phase 1/4-approximation guarantee. They also give a non-monotone Frank-Wolfe variant with 1/e-approximation guarantee. Experiments, however, suggest that the former (two-phase) is always comparable or better than the latter in practice. The analysis in Subsection 3.2 also simplifies previous work [7, 20] by a different proof. Overall, I find the technical contributions worthy of acceptance.

Reviewer 2



This paper presents a new approach for non-monotone DR sub modular optimisation. The strength of this paper over existing approaches is proposing a solution to a more general problem setting than having an approach subject to constraints such as the box constraints. The proofs are well constructed and presented. However the experimental results are somehow very limited. The parameters used for the simulated data are very limited and not sure if the authors have performed a much more simulations with different parameters to assess more in depth the "performance" of their approach over the existing ones. Also, for the real data set example, it is not very clear if that the proposed two-phase Frank-Wolfe algorithm performs much better than the others. From the figures, it seems performing just slightly better but not very convincing. For real-world problems, it is typically difficult to verify whether the assumptions hold or not, so a comparison with the DoubleGreedy algorithm would have been appropriate to see how much gain we have by considering a more general constraint.

Reviewer 3



The paper proposes two algorithms with approximation guarantee and convergence rate on non-monotone continuous DR-submodular maximization under general down-closed convex constraint, which extends previous results on "box" constraints. Experiment results show the proposed algorithms have better performance compared to projected gradient on both synthetic and real-world data. The paper also generalizes the problem to a conic lattice setting, and shows reduction to continuous submodular optimization. The paper is theoretically concrete. The properties of constrained DR-submodular maximization discussed in the paper, especially the local-global relation could be intriguing for future research. Thanks for the feedback. I think if the authors can come up with more applications of the proposed constrained DR-submodular optimization, the contribution of the work could be improved significantly. Minor: line 460 (supplement) "show" instead of "shwo"